# HPV Infection in Middle Ear Squamous Cell Carcinoma: Prevalence, Genotyping and Prognostic Impact

**DOI:** 10.3390/jcm10040738

**Published:** 2021-02-12

**Authors:** Giovanni Cristalli, Aldo Venuti, Fabiola Giudici, Francesca Paolini, Fabio Ferreli, Giuseppe Mercante, Giuseppe Spriano, Francesca Boscolo Nata

**Affiliations:** 1Otorhinolaryngology Unit, Ospedali Riuniti Padova Sud “Madre Teresa di Calcutta” Hospital, 35043 Monselice, Italy; giovanni.cristalli@gmail.com; 2HPV Unit, UOSD Tumor Immunology and Immunotherapy, IRCCS Regina Elena National Cancer Institute, via Elio Chianesi 53, 00144 Rome, Italy; aldo.venuti@ifo.gov.it (A.V.); francesca.paolini@ifo.gov.it (F.P.); 3Unit of Biostatistics, Epidemiology and Public Health, Department of Cardiac, Thoracic, Vascular Sciences and Public Health, University of Padua, 35122 Padua, Italy; fgiudici@units.it; 4Department of Biomedical Sciences, Humanitas University, via Rita Levi Montalcini 4, 20090 Pieve Emanuele, Italy; fabio_ferreli@yahoo.it (F.F.); mercante.giuseppe@gmail.com (G.M.); bspriano@email.it (G.S.); 5Otorhinolaringology—Head and Neck Surgery Unit, IRCCS Humanitas Research Hospital, via Manzoni 56, 20089 Rozzano, Italy

**Keywords:** HPV, p16, middle ear squamous cell carcinoma, prognosis, recurrence

## Abstract

Middle ear squamous cell carcinoma (MESCC) is rare. Human Papilloma Virus (HPV) infection has been found in a significant number of cases of MESCC. Despite the emerging role of HPV in oncogenesis, its role in the pathogenesis and prognosis of MESCC is not known. This study aims to identify the prognostic impact of alpha and beta HPV in MESCC and its correlation with p16 protein. We retrospectively investigated 33 patients with MESCC surgically treated between 2004 and 2016. HPV DNA was ascertained by polymerase chain reaction (PCR) and P16INK4a detection was performed. Disease-specific survival (DSS) and cumulative incidence of recurrence were calculated in relation to HPV presence and genotype. p16 sensitivity, specificity, positive predictive value (PPV), and negative predictive value (NPV) in predicting HPV infection were calculated. HPV was detected in 66.7% of patients (36.4% alpha HPV, 63.6% beta HPV). Five-year DSS was 55.0% and was not statistically related to HPV presence (*p* = 0.55) or genotype (*p* = 0.87). Five-year cumulative incidence of recurrence was 46 %, and was not statistically related to HPV presence (*p* = 0.22) or genotype (*p* = 0.44). p16 sensitivity, specificity, PPV, and NPV in predicting HPV infection were 27.3%, 36.4%, 46.2%, and 20.0%, respectively. In our experience, beta HPV was more frequent than alpha HPV in MESCC. Neither HPV presence nor HPV genotypes relate to DSS or cumulative incidence of recurrence. p16 expression was not predictive for HPV infection in MESCC. The role of HPV infection in oncogenesis, maintenance, and prognosis of MESCC seems to be different from that in oropharynx and skin cancer.

## 1. Introduction

Primary middle ear squamous cell carcinoma (MESCC) has an annual incidence of 0.8–1 per million inhabitants and it is the most common histology in this site [1]. Risk factors such as ultraviolet (UV) rays and contact with carcinogens are not relevant in MESCC, compared to squamous cell carcinoma (SCC) of the pinna. Conversely, MESCC could be associated to chronic otitis media, with or without cholesteatoma as well as exposure to ionizing radiations [1,2]. In particular, the association between cancer and inflammation is not new: it was firstly hypothesized in the nineteenth century after the observation that tumors often arose at sites of chronic inflammation, and that inflammatory cells were present in biopsied samples from tumors. Following studies have confirmed this association, that nowadays represents a field of interest. The triggers of chronic inflammation could be both microbial or autoimmune [3]. Differently from normal inflammation, chronic inflammation is not self-limiting: consequently, it induces chronic exposure to cytokines and reactive oxygen species that increase cell proliferation, mutagenesis, oncogene activation, and angiogenesis [4].

The carcinogenic role of Human Papilloma Virus (HPV) has emerged in last decades. Alpha HPVs can be categorized as high-risk (HR) or low-risk (LR) according to their carcinogenic potential [5], whereas some types of beta HPV have been associated with cutaneous cancer [6]. 

Beta HPVs have been detected also in oral cancer and in middle ear tumors, but their role in carcinogenesis has not been fully investigated [7,8,9].

HPV has been found both in inflammatory middle ear disease and in cancer, but its role as co-pathogen is not fully understood yet [1]. Furthermore, the role of beta HPV is becoming evident as co-carcinogenic factor [6].

The prognostic role of HPV infection, using p16 as a diagnostic biomarker, has been demonstrated in oropharyngeal SCC (OPSCC) [10]; conversely, the prognostic significance of HPV infection and its correlation with p16 protein expression have been less investigated in MESCC. 

The objective of the present study was to evaluate the association between HPV infection and MESCC, its prognostic role, and possible association with p16 expression.

## 2. Materials and Methods

### 2.1. Patients

Patients diagnosed with MESCC who underwent surgery between 2004 and 2016 were selected from our institutional oncologic database to be retrospectively reviewed. The study was conducted according to the principles stated in the Declaration of Helsinki and approved by the local Ethics Committee (# RS852010). Patients gave their written informed consent for the use of their data. Inclusion criteria were: primary tumors of the middle ear with SCC histology, surgical resectability, and at least a 24-months follow-up. Exclusion criteria were: patients with recurrent tumors, skin tumors extended to the middle ear, previous radiotherapy (RT), and/or chemotherapy.

### 2.2. Tumor Assessment and Treatment 

Tumor staging was determined using ear and neck computed tomography (CT) and/or magnetic resonance imaging (MRI) with contrast, and a chest CT. Tumors were classified according to the Arriaga staging system [11]. Surgical treatment, reconstruction, and adjuvant therapy were recorded. Partial/sub-total parotidectomy was performed to obtain a wider surgical field and to better visualize the facial nerve. Therapeutic or elective neck dissection (ND) associated with dissection of the retroauricular and occipital nodes was performed in case of clinical or radiologic positive neck or advanced tumor stage (III or IV). 

Indications for adjuvant intensity-modulated radiotherapy (IMRT) were Arriaga stage ≥III or close/positive resection margins at histology [12]. IMRT was administered five days per week; a total dose between 60–70 Gy in 35 fractions was used. Patients with three or more neck node metastases or extra capsular-extensions also received chemotherapy (Cisplatin and 5-Fluorouracil).

The presence of HPV and EBV (to exclude contamination from the nasopharynx) DNA was retrospectively ascertained on histological samples. Surgical specimens were embedded in formalin and a nested polymerase chain reaction (PCR) with consensus primers was carried out followed by direct sequencing according to previously reported standard procedures [8]. P16INK4a detection was performed with a commercial kit (CINtec^®^ p16INK4a, Roche Diagnostics, Basel, Switzerland) and was defined by a strong and diffuse nuclear and cytoplasmic staining in ≥70% of the tumor cells. 

### 2.3. Follow-Up

Follow-up clinical evaluations were scheduled every three months for the first two years and then twice a year for a total of at least five years. Head and neck CT scan or MRI were performed every 6 months for the first two years and thereafter once a year for the following years to identify loco-regional recurrence. 18-fluorodeoxyglucose (FDG) positron emission tomography (PET) CT scan was performed once a year to detect loco-regional and/or distant metastasis.

### 2.4. Statistical Analysis

Categorical variables were reported as absolute frequencies and percentages. Continuous variables were summarized as mean ± standard deviation and range. Disease-specific survival (DSS) rate was defined as the interval between the date of surgery and the date of the death by the specified cancer, with censoring at the last examination date for patients who were alive. The median follow-up was computed for censored patients, excluding patients with events of interest (reverse Kaplan–Meier method). DSS was estimated using the Kaplan–Meier approach and differences of survival between groups (HPV positive vs. HPV negative, alpha HPV vs. beta HPV) were assessed through the Log-rank test.

Disease-free survival (DFS) rate was defined as the interval between the date of surgery and the date of local (T) and/or regional (N) recurrence, with censoring on the last examination date for patients who were disease-free. The cumulative incidence function (CIF) was used to describe recurrence trends, taking into account the presence of competing risks (i.e., death) since most patients died precluding the occurrence of relapse. Cumulative incidences of recurrence between groups (HPV negative vs. HPV positive, alpha HPV vs. beta HPV) were compared with the Gray test. CIF was calculated with the R package cmprsk [13]. 

The performance of p16 to predict HPV infection calculating sensitivity, specificity, positive predictive value (PPV), and negative predictive value (NPV) was evaluated for all types of HPV and for alpha versus beta HPV with 95% CI.

## 3. Results

A total of 33 patients met the inclusion criteria and were recruited for the present study. Mean age was 75 ± 11 (range, 40–91). Figure 1 shows an example of preoperative CT (Figure 1a,b), intraoperative view (Figure 1c) and 1-year follow-up (Figure 1d).

Information on patient demographics, tumor characteristics, and treatment type are summarized in Table 1. 

HPV DNA was found in 66.7% of MESCC, of which 36.4% were alpha HPV and 63.6% were beta HPV. Eighteen patients died from MESCC, 14 died from other causes, while 1 patient was alive at the time of follow-up closure. 

Median follow-up was 9.62 years (IQR 4.50–11.14), (among the 18 patients who died for specific causes, 13 died before 5 years), and the median survival was 6.06 years. Five-year DSS was 55.0% (CI 95%: 35.2–71.0%); DSS was not statistically associated to HPV presence (*p* = 0.55) (Figure 2a) nor to HPV genotype (*p* = 0.87) (Figure 2b).

The 5-year cumulative incidence of MESCC recurrence was 46.0% (CI 95%: 28–62%) with a median time of MESCC recurrence at 5.30 years. Among the 19 patients who experienced MESCC recurrence, 10 (52.6%) had T recurrence, and 9 (47.4%) had N recurrence. HPV-negative patients experienced a higher incidence of recurrence compared to HPV-positive patients (0.64% versus 0.36%), but the difference was not statistically significant (*p* = 0.22) (Figure 3a). Recurrence risk did not relate to HPV genotype (*p* = 0.44) (Figure 3b).

Sensitivity, specificity, PPV, and NPV of p16 in predicting HPV infection were 27.3% (CI 95%: 10.7–50.2%), 36.4% (CI 95%: 10.9–69.2%), 46.2% (CI 95%: 19.2–74.9%), and 20.0% (CI 95%: 5.7–43.7%), respectively.

Sensitivity, specificity, PPV, and NPV of p16 in predicting alpha HPV infection were 37.5% (CI 95%: 8.5–75.5%), 36.4% (CI 95%: 10.9–69.2%), 30.0% (CI 95%: 6.7–65.2%), and 44.4% (CI 95%: 13.7–78.8%), respectively.

Sensitivity, specificity, PPV, and NPV of p16 in predicting beta HPV infection were 21.4% (CI 95%: 4.7–50.8%), 36.4% (CI 95%: 10.9–69.2%), 30.0% (CI 95%: 6.7–65.2%), and 26.7% (CI 95%: 7.8–55.1%), respectively.

## 4. Discussion

This study focused on the prognostic impact of HPV in MESCC, and the relationship between p16 presence and HPV infection status in a homogeneous group of patients in terms of staging, treatment, and follow-up. 

The presence of alpha and beta HPV is reported in middle ear mucosa but their role in carcinogenesis, and the relationship with middle ear disease, still needs to be established [14]. Alpha HPV can be found both in skin and mucosa [15], while beta HPV is very commonly isolated on the skin and eyebrows. Notably, beta HPV was present in 63.6% of cases in our series of MESSC. This data confirms that beta HPV genotype can have both mucosal and cutaneous tropism, moreover, this species shares biological similarities with HR HPV [16,17,18]. A significant association between beta HPV and cancer has been demonstrated in the head and neck region [17,19]: moreover, the ability of beta HPV E6 and E7 proteins to transform epithelial cells in vitro and in vivo models has been highlighted [18]. In addition, the ability of HPV to elude cellular-mediated immunity associated with chronic otitis media could represent a risk factor for the development of SCC [20]. Otorrhea was present in 66.7% of our sample; anyway, it was not possible to determine if this sign preceded the development of MESCC or if it was a consequence of a suprainfection in case of tumor infiltration of the external ear. Future studies may clarify this issue.

The difference between HPV-positive MESCC in our cohort (66.7%) compared to that reported by Surono et al. (29%) may be due to the fact that HPV was detected by immunohistochemistry, a low-sensitive method, in the latter [21]. Previous studies focused on HR alpha HPV in MESCC, with a reported prevalence of 89% [22] and 78.6% [1]. These high percentages of HPV positivity compared to our results (36.4%) could be explained by the different technologies/primers utilized and geographical areas [23]. 

The prognostic impact of HPV is well known in pharyngeal SCC. Specifically, HPV-positive OPSCC are characterized by a better progression-free survival and overall survival [24,25], and lower cancer-specific mortality [26]. Better locoregional control, DFS, and overall survival in HPV-positive compared to HPV-negative nasopharyngeal cancer have also been demonstrated, but this difference did not reach statistical significance [24]. Similarly, a statistically significant difference in DSS according to HPV status had not been found in patients with SCC of the temporal bone, even if survival in the HPV-positive group was higher [27]. Our data did not shown any association between HPV status and DSS; furthermore, DSS was still not influenced by HPV presence when comparing alpha versus beta HPV genotypes. HPV-negative patients were characterized by a higher incidence of recurrence, but a significant difference on the cumulative risk of recurrence according to HPV presence and genotype was not demonstrated in our study. 

Previous studies reported the presence of p16 in 38–64% of oropharyngeal, 7–28% of hypopharyngeal, and 9–80% of nasopharyngeal mucosal cancers [24]. In our cohort, p16 positivity was found in 39.4% of patients with MESCC.

Alpha HPV can induce up-regulation of p16 and concordance between p16 and HPV was the basis for the current use of p16 as a surrogate immune-histochemical (IHC) marker for HPV infection in OPSCC [28]. Conversely, in our experience, p16 was not predictive for HPV infection in MESCC, as reported for hypopharyngeal and nasopharyngeal cancers [24] and in geographical areas where p16 does not seem to be a surrogate marker of HPV positivity in oral cavity and laryngeal carcinoma [29]. Thus, p16 expression might be dependent on other non-HPV causes. Senescence, reactive oxygen species, and DNA damage due to chronic inflammation should be suspected as possible factors in p16 expression in MESCC [30]. Indeed, up to 20% of cancers have been linked to chronic infection. Specifically, the production of reactive oxygen and nitrogen species, usually produced to fight infection, is persistent during a chronic infection, thus inducing DNA damage [4].

## 5. Conclusions

HPV can be present in MESCC. In our sample, MESCC prognosis was not influenced by the presence nor by the HPV genotype; moreover, p16 positivity was not associated with HPV presence. A high percentage of MESCC patients presented with otorrhea. Future studies should investigate on the role of chronic infection in MESCC pathogenesis. Considering the low incidence of this cancer, multicenter studies are advised in order to verify this hypothesis.

## Figures and Tables

**Figure 1 jcm-10-00738-f001:**
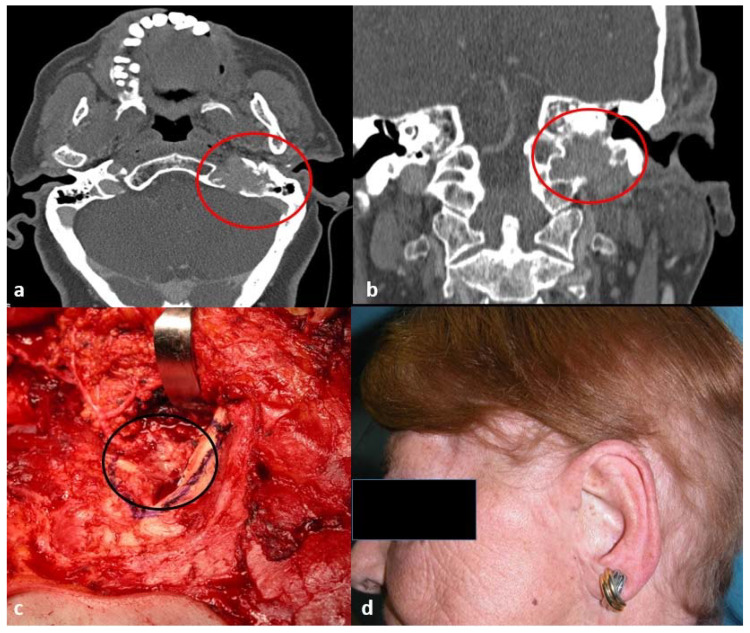
Preoperative, intraoperative, and post-operative images. (**a**) Preoperative computed tomography (CT) scan axial view; the red circle indicates the tumor; (**b**) preoperative CT scan coronal view; the red circle indicates the tumor; (**c**) intraoperative view, the black circle indicates the tumor; (**d**) 1-year post-operative clinical evaluation.

**Figure 2 jcm-10-00738-f002:**
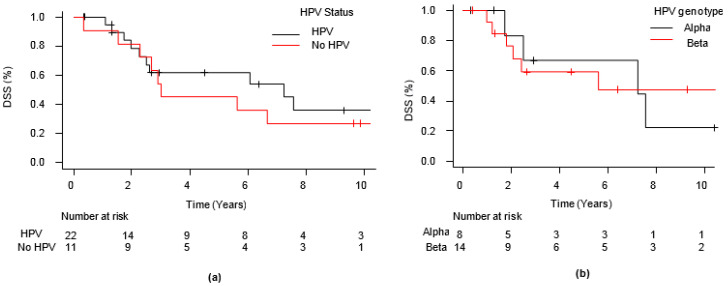
Five-year disease-specific survival. (**a**) Five-year disease-specific survival (DSS) rate in relation to Human Papilloma Virus (HPV) presence; (**b**) five-year DSS rate in relation to HPV genotype.

**Figure 3 jcm-10-00738-f003:**
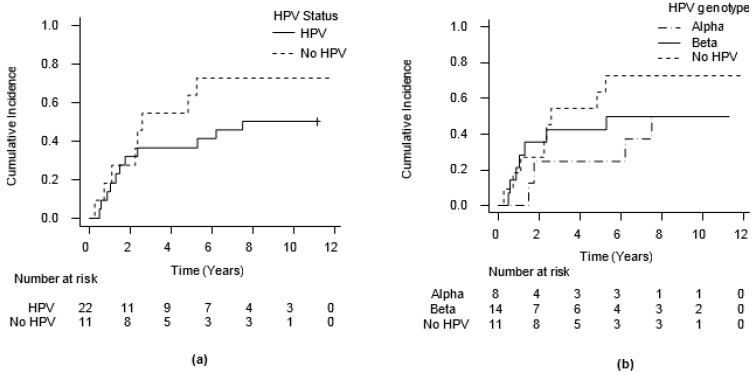
Cumulative incidence of recurrence. (**a**) Cumulative incidence of recurrence in relation to HPV presence, using competing risks (CR) method; (**b**) cumulative incidence of recurrence in relation to HPV genotype, using CR method.

**Table 1 jcm-10-00738-t001:** Patients demographics, tumor and treatment characteristics.

Variables	N (%)
Patients	33
Male	25 (75.8)
Aural discharge	22 (66.7%)
**T stage Arriaga**	
T2	5 (15.2)
T3	28 (84.8)
**N stage Arriaga**	
N0	19 (57.6)
N1	3 (9.1)
N2b	11 (33.3)
**Staging Arriaga**	
II	5 (15.2)
III	12 (36.4)
IV	16 (48.4)
**HPV positivity**	22 (66.7)
**Genotypes**	
**Alpha**	8 (36.4)
16	3
18	5
**Beta**	14 (63.6)
8	1
14	1
20	1
23	1
25	2
98	1
99	1
100	1
113	1
120	1
129	1
150	1
159	1
**p16 positivity**	13 (39.4)
**EBV negativity**	33 (100)
**Surgical treatment**	
Local resection	2 (6.0)
Lateral temporal bone resection	22 (66.7)
Subtotal temporal bone resection	9 (27.3)
**Reconstructive flap**	
None	6 (18.2)
Pectoralis major myo-cutaneous pedicle flap	15 (45.5)
Temporalis pedicle flap	6 (18.2)
Skin graft	2 (6)
Forearm microvascular free flap	2 (6)
Antero-lateral thigh free flap	2 (6)
**Neck dissection**	
Elective	19 (57.6)
Therapeutic	14 (42.4)
**Partial/Subtotal parotidectomy**	32 (97.0)
**Parotid infiltration at histology**	12 (36.4)
**Adjuvant treatment**	
RT	26 (78.8)
CT	12 (34.4)

## Data Availability

The data presented in this study are available on request from the corresponding author. The data are not publicly available due to privacy restrictions.

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
