# Peer review of "HPV Infection in Middle Ear Squamous Cell Carcinoma: Prevalence, Genotyping and Prognostic Impact"

_jcm, 2021, doi:10.3390/jcm10040738_

Round 1
Reviewer 1 Report
- well described Introduction chapter including the role of HPV infection in SCC development.
- well described tumor histopathology and treatment with follow-up
- Results are well presented with tables and diagrams with follow up. The miost imortant is the result concerning p16 expression and HPV innfection in SCC.
- Dissscussion chapter is based on the newest data with clear conclusions.
Author Response
We thank the reviewer for his/her positive comments.
English has been checked for grammar and style.
Reviewer 2 Report
This is a retrospective study of a small cohort of 33 cases with MESCC, however the methodology to study HPV prevalence and recurrence in relation to HPV genotype of MESCC is correct. The introduction, results and discussion are clearly presented.
Author Response
We agree with the reviewer that the conclusions about the role of HPV infection in oncogenesis, maintenance and prognosis, and the possible impact of chronic infection on SCC could be only postulated at with the present result, and that future research is needed to fully understand these associations. Conclusions have been changed accordingly, both in the abstract (line 37) and text (lines 232-240)
English has been checked for grammar and style.
Reviewer 3 Report
The article refers to a relatively rare pathology with therefore not numerous cases probably collected in different structures. It is not clear how many patients died from causes other than the disease. It is not specified in the introduction how many patients had a follow-up of less than the 5 years foreseen for the calculation of the 5 - years free DSS
Author Response
We thank the reviewer for this observation.
We added in the Results section (see line 143) that 14 patients died for other causes. Moreover, data about follow up have been clarified in the Results section (see text lines 145-146). Median follow up was 9.62 years because, as stated in the statistical analysis section, it was computed for censored patients, excluding patients with events of interest (death for specific cause). As regards your question, in total 20 patients had a follow up of less than 5 years: specifically they were 13 patients among those experiencing the event (death for specific cause), and 7 among patients without the event.
More details about the association between chronic inflammation and cancer have been added in the introduction section (see text lines 47-54)
More details have been added in the results (see lines 141-148, 154-156)
Reviewer 4 Report
This is an interesting article with a relatively high number of patients with the uncommon middle ear SCC. The findings are perhaps not unexpected with the relatively high HPV prevalence in upper airway SCCs and the concept of the unified airway. Perhaps the most interesting finding which warrants further study is the 66.7% correlation with chronic otorrhoea. However one must question whether the otorrhoea preceded the development of the SCC or is a consequence of the tumour open to the external ear and secondarily infected.
Perhaps as you surmise CSOM with a promoter such as HPV may be a significant risk factor for middle ear squamous cell carcinoma.
Author Response
We agree with the reviewer that the association with chronic otitis is the most interesting result of the present study. As underlined by the reviewer, and now added in the text (see lines 189-193), considering the retrospective nature of the study and the high number of patients who died, it was not possible in the present study to define if aural discharge preceded the development of SCC or if it was the cancer itself to invade the external ear, to infect, and to cause aural discharge. An ongoing study at our department is trying to clarify this unanswered question, but the number of patients to be included in order to obtain significant result is high and, as known, this cancer is rare.
English has been checked for grammar and style.